# Global Cropland Expansion Enhances Cropping Potential and Reduce its Inequality among Countries

Xiaoxuan Liu[1,2], Peng Zhu[3,4], Shu Liu[2], Le Yu[2,5,6*], Yong Wang[2], Zhenrong Du[7], Dailiang Peng[1], Ece Aksoy[8], Hui Lu[2] & Peng Gong[4,9]

[1]Aerospace Information Research Institute, Chinese Academy of Sciences, Beijing, 100190, China
[2]Department of Earth System Science, Ministry of Education Key Laboratory for Earth System Modeling, Institute for Global Change Studies, Tsinghua University, Beijing, 100084, China
[3]Department of Geography, The University of Hong Kong, Hong Kong 999077, China
[4]Institute for Climate and Carbon Neutrality, The University of Hong Kong, Hong Kong 999077, China
[5]Ministry of Education Ecological Field Station for East Asia Migratory Birds, Tsinghua University, Beijing 100084, China
[6]Tsinghua University (Department of Earth System Science)- Xi'an Institute of Surveying and Mapping Joint Research Center for Next-Generation Smart Mapping, Beijing 100084, China
[7]School of Information and Communication Engineering, Dalian University of Technology, Dalian 116024, China
[8]Geospatial Unit, Food and Agriculture Organization of the United Nations, Rome 00153, Italy
[9]Department of Geography, Department of Earth Sciences and Institute for Climate and Carbon Neutrality, University of Hong Kong, Hong Kong, Hong Kong 999077, China

*Correspondence to*: Le Yu (leyu@tsinghua.edu.cn)

**Abstract.** Global cropland expansion has been recognized as a key driver of food security. However, cropland expansion induced alterations in biophysical properties of the Earth's surface and greenhouse gas emissions may potentially impact the Earth's climate system. These changes could, in turn, affect cropland productivity and the potential distribution of croplands, although the underlying mechanisms remain relatively underexplored. In this study, a global climate model was employed to quantify the impact of global cropland expansion on cropping potential, utilizing observed and derived cropland expansion data. Our findings reveal that since 10000 BC, a 28% increase in cropland expansion has led to a 1.2% enhancement in global cropping potential, owing to more favorable precipitation and temperature conditions. This suggests that global cropland expansion yields dual benefits to crop production. However, in regions with low growth rates of cropping potential, cropland expansion proves to be an inefficient method for augmenting local crop potential yield. As croplands continue to expand worldwide, the capacity to support populations in different regions is altered, thereby reducing cropping potential inequality among nations.

## 1 Introduction

Land use change is the fundamental result of human activities that disturb the Earth's surface, and it plays a crucial role in global change (Foley et al., 2005; Rockström et al., 2009; Sasmito et al., 2019). This change in land use is closely connected to climate change, ecological environmental change, the sustainable utilization of natural resources, and human health (Hasan,

Zhen, Miah, Ahamed, & Samie, 2020; Kates et al., 2001; Wall, Nielsen, & Six, 2015). The growth of population and economic development has generated an increasing demand for land commodities, which accelerating a continuous expansion of global agricultural land through activities such as deforestation, reclamation of grasslands, and the conversion of cropland from lakes (Foley et al., 2005; Zabel et al., 2019). At present, approximately 40% of the Earth's ice-free land surface is used for agricultural activities, totaling over 1500 million hectares (Mha) of cropland (Potapov et al., 2022; Ramankutty, Evan, Monfreda, & Foley, 2008; Ritchie & Roser, 2013; Yu et al., 2013). Moreover, the expansion of cropland is projected to continue under all scenarios (Hurtt et al., 2020; Xiaoping Liu et al., 2017), emphasizing the importance of such expansion in ensuring food security and agricultural production in response to rapid population growth (Delzeit, Zabel, Meyer, & Václavík, 2017; Levers, Butsic, Verburg, Mueller, & Kuemmerle, 2016).

While existing research primarily focuses on the effects of cropland expansion on soil degradation, climate change, and biodiversity loss (Ortiz-Bobea, Ault, Carrillo, Chambers, & Lobell, 2021; Searchinger et al., 2015), limited knowledge exists regarding the repercussions of this expansion on the croplands themselves. Previous studies examining the impacts of historical cropland expansion have often been complicated by land use changes unrelated to agricultural expansion and contraction (Lawrence & Chase, 2007; Sterling, Ducharne, & Polcher, 2013). Several studies have analyzed the effects of land cover changes on climate change. For example, Lawrence et al. used MODIS data to our understanding of land cover changes and found that changes in soil cover led to local warming and drought near the ground (Lawrence & Chase, 2007); Yan et al. studied climate change caused by land cover from 1 AD to 2000 AD using an earth system model and found that an increase in cropland and a decrease in forests corresponded to a downward trend in the global average annual temperature (Yan, Liu, & Wang, 2017); Conversely, Sampaio et al. discovered that large-scale deforestation resulting from rangeland and soybean expansion in the Amazon significantly alters regional climate, leading to warming and dryness after deforestation (Sampaio et al., 2007); Additionally, Arora et al., using climate models combined with the carbon cycle, found that if all or half of the world's agricultural land were allowed to revert to forest, temperatures would be 0.45°C and 0.25°C lower by 2100 compared to scenarios in which no reversion occurs (Arora & Montenegro, 2011). However, these studies generally consider changes in all land types, and thus, the specific influence of cropland expansion on climate change remains under-quantified.

Furthermore, cropland expansion often involves the encroachment of agricultural activities on other land types, such as deforested areas and grasslands (Bahar et al., 2020; Xiaoxuan Liu et al., 2018). However, due to the ongoing depletion of natural resources, sustainable intensification of cropland has emerged as a means of feeding a growing population while ensuring both human and environmental well-being (Cole, Augustin, Robertson, & Manners, 2018; Godfray et al., 2010). This approach involves practices such as multiple cropping, fertilization, irrigation, and agricultural mechanization, all aimed at enhancing agricultural productivity within existing cropland (Mauser et al., 2015). These sustainable approaches are expected to play a substantial role in increasing future food production as more effective strategies (Folberth et al., 2020; W.-B. Wu et al., 2014).

The dynamic changes in cropland, including its status and cropping potential, have significant impacts on global climate change, terrestrial ecosystems, biogeochemical cycle processes, and global land-ocean interactions (Bennett, Carpenter, &

Caraco, 2001; DeFries, Foley, & Asner, 2004; Steinfeld et al., 2006; Zhao et al., 2019). Research on the potential of cropland is therefore of great importance in understanding and responding to changes in regional and global ecological environments (Hu et al., 2020). It can help fully explore the potential of cropland based on its existing extent, provide better guidance for land use planning, facilitate adjustments in agricultural structure, and coordinate grain trade (Mehrabi, Ellis, & Ramankutty, 2018; Parodi et al., 2018). Moreover, changes in the spatial distribution of cropland can affect global climate change through atmospheric circulation, which in turn further affects cropland itself (Bonan, Pollard, & Thompson, 1992; Brovkin et al., 2004; Iizumi & Ramankutty, 2015; Yang et al., 2015). Quantifying and understanding the impact of global cropland dynamics on the climate production potential of cropland can assist decision-makers in considering the sustainability of actions within evolving cropland landscapes (Fischer, Shah, N. Tubiello, & Van Velhuizen, 2005; J. Wang, Vanga, Saxena, Orsat, & Raghavan, 2018). This, in turn, can contribute to sustainable cropland development, serve the goals of sustainable agricultural development and food security, and contribute to the achievement of the United Nations Sustainable Development Goals (SDGs) (UN, 2018).

In this study, we utilized an Earth system model to assess the influence of cropland expansion-induced climate change on global cropping potential. Our model simulation incorporated observed and derived data on cropland expansion and contraction from 10,000 BC to 2015, while keeping other land categories unaffected by these changes. Through Earth system model experiments, we examined the resulting temperature and precipitation conditions to assess the impact of global cropland expansion on total cropping potential using a cropping potential model. Subsequently, we analyzed regional changes in cropping potential and formulated specific policy recommendations for different geographical regions.

## 2 Methods

**Cropland Expansion datasets from HYDE.** The History Database of the Global Environment (HYDE version 3.2) datasets are used to extract the cropland expansion distribution from 10000BC to 2015 (Goldewijk, Beusen, Doelman, & Stehfest, 2017). For further use in our research, multiple land types in HYDE datasets are divided into six general types (i.e., bareland, cropland, grassland, ice/snow, urban area, and woodland), according to a general classification system of Finer Resolution Observation and Monitoring - Global Land Cover (FROM-GLC) based on Gong et al. (Gong et al., 2013) and Liu et al. (Xiaoxuan Liu et al., 2018), see Tab. S2. Considering the HYDE datasets have no clear-cut distinctions between semi-natural treeless, wild, remote-treeless, and barren lands, a global potential vegetation dataset (PV) is introduced to discriminate the 10000BC land cover. All the cropland expansion datasets for 10000BC, 1850, 1990, and 2015 are shown in Fig. S1, representing the land cover distribution derived from HYDE and PV. To ensure that only cropland expansion is retained in this research, we only keep areas with land cover changes in cropland and other areas of variation are set the same as the base year, 2015 (see the example at Fig. S2). Then, to fit the resolution of the following climate model at ~ 2°, we aggregated the original 1 km resolution into 2°×2° grid cells and calculated the area proportions of every land cover change type in each 2° grid cell (see the example at Fig. S3).

**Global Climate Model.** The global climate model (GCM) used in this study is the National Center for Atmospheric Research (NCAR) Community Earth System Model version 1.2.1 (CESM1.2.1), including atmosphere, land, ocean, and sea ice. The most relevant components for this study are the atmosphere and land components represented by the Community Atmosphere Model version 5.3 (CAM5.3) and the Community Land Model version 4 (CLM4), respectively. The global climate model used

in this study is the National Center for Atmospheric Research (NCAR) Community Earth System Model version 1.2.1 (CESM1.2.1), including atmosphere, land, ocean, and sea ice component models. Among them, the component of the atmosphere and land are more relevant for this study. The reliability of the CESM has been confirmed in numerous previous studies, making it suitable for applications such as climate change simulation and climate model analysis (Hurrell et al., 2013; Kay et al., 2015). CAM5.3, which uses Finite-Volume dynamical core and a suite of parameterization schemes for representing

various atmospheric physical and chemical processes (Neale et al., 2010). In CLM4, including multiple land surface processes (Oleson et al., 2010), spatial land surface heterogeneity is represented as fractional coverages of multiple land types coexisting in each grid cell. The four cropland expansion datasets (10000BC, 1850, 1990, and 2015) derived from HYDE are further handled to follow the land cover classification of the CLM4. Each cropland expansion dataset includes six types of land cover. The woodland land type could be further disaggregated into forest and shrubland according to the mean annual proportion of

the European Space Agency Climate Change Initiative Land Cover (ESA CCI-LC) dataset (P Defourny et al., 2017; Pierre Defourny et al., 2009). The fractional coverage of bareland, cropland, ice/snow, and urban coverage types can be directly applied to the model. At the same time, those of the other land cover types need to be divided into corresponding subtypes in CLM4 according to their relative ratios (S. Liu et al., 2021) (Fig. S4-7).

**Experimental design.** This study focuses on the climatic effects of the external forcing of cropland expansion. Therefore, four simulation groups are conducted: crop10000BC, crop1850, crop1990, and crop2015 which are driven by different land surface data of 10000BC, 1850, 1990, and 2015, respectively. These simulation experiments, only differing in cropland cover, were ran in the Atmospheric Model Intercomparison Project (AMIP)-type, using fully prognostic atmosphere and land models with prescribed, seasonally varying present-day climatological (1981-2001 mean) sea surface temperatures and sea ice

concentrations (Hurrell et al., 2013). Other external forcings such as solar radiation, anthropogenic aerosol emissions, and greenhouse gas concentrations were also fixed in the present-day climatological conditions (Eyring et al., 2016). This is because taking into account the nonlinear responses of climate to external forcings (Rohrschneider, Stevens, & Mauritsen, 2019), employing various external forcing levels would inevitably perturb the signal of the external forcing of cropland expansion. Each simulation group has 10 simulations differing in their initial conditions by adding a random round-off level

(order of $10^{-14}$ K) error to the initial air temperature fields, and simulations are run for 15 years at a horizontal resolution of ~ 2°. The first ten years are regarded as spin-up, the last five years of daily output are used for further analysis.

**Diagnosis of surface air temperature and precipitation changes.** Surface air temperature changes induced by cropland expansion are diagnosed by the thermodynamic energy equation in the pressure coordinate (Chemke et al. 2016; Lee et al. 2011), given by Eq.(1):

$$\delta\bar{T} \approx \gamma^{-1}\left(-\delta\overline{(\vec{V_h}\cdot\nabla_h T)} + \delta\overline{(S_p\omega)} + \delta\overline{Q_s} + \delta\overline{Q_{ld}} + \delta\overline{F_{sh}} + \delta\overline{Q_q}\right) \quad (1)$$

where $\delta$ denotes the differences between simulation experiments (here crop2015 minus crop10000BC, crop1850, and crop1990, respectively) and the overbar denotes the mean time. T is the surface air temperature, and the right-hand side of the equation shows respective contributions to its changes. The first term is the horizontal temperature advection, in which Vh is horizontal wind. The second is adiabatic warming/cooling, where Sp is the atmosphere stability parameter and $\omega$ is the vertical velocity in the pressure coordinate. The remaining four terms represent shortwave radiative heating rate, surface downward longwave flux, surface sensible heat flux, and latent heat release plus vertical diffusion. The units of the six terms on the right-hand side are unified to W/m² by multiplying the specific heat capacity of air and the air mass per unit area at the near-surface (Chemke, Kaspi, & Halevy, 2016). The influence of surface sensible and latent heat fluxes on air temperature changes encompasses the impacts of numerous biophysical factors, including surface albedo, surface emissivity, aerodynamic resistance, and surface resistance (Lee et al., 2011; Chen et al., 2020). This is because these biophysical factors directly affect surface temperature, thereby indirectly influencing near-surface air temperature through the near-surface turbulence of sensible and latent heat (Li, Piao, Chen, Ciais, & Li, 2020; Zeng et al., 2017).

Precipitation changes induced by cropland expansion are examined by the atmospheric moisture budget equation (Y. Wang, Zhang, & Jiang, 2018), given by Eq.(2):

$$\delta\bar{P} \approx \delta\overline{(-W\nabla\cdot\vec{V})} + \delta\overline{(-\vec{V}\cdot\nabla W)} + \delta\bar{E} \quad (2)$$

where $P$ denotes precipitation, $W$ is the column-integrated precipitable water, V represents the total horizontal moisture transport normalized by the $W$, and $E$ is the evaporation. Precipitation changes can be disentangled from moisture convergence, moisture advection, and evaporation on the right-hand side.

**Bias correction of model simulations.** We use the widely applied 'delta method' for climate model bias correction (Diffenbaugh & Burke, 2019), in which model-simulated changes are applied to observations. AgERA5 (Agrometeorological ECMWF Re-Analysis v5, Agrometeorological European Centre for Medium-Range Weather Forecasts Re-Analysis v5) is used for corrections of surface air temperature and precipitation from the simulations. This dataset provides daily surface meteorological data from 1979 to the present as input for agriculture and agroecological studies. The averaged values from 2010 to 2019 of AgERA5 replace the simulation results of the crop2015 simulation. Those of the other three experiments are calculated as the ERA5-Agro added by their respective differences from the crop2015. Moreover, we verified the correctness of the corrected temperature and precipitation by analysing spatial distributions and PDF (Probability Density Function) (Fig. S8-11). The results show that the CESM can well capture the spatial distributions and PDFs of temperature and precipitation in ERA5.

**Climate cropping potential intensity model.** Climate cropping potential denotes the utmost capacity for multi-cropping achievable after thorough climate resource assessment. The climatological precipitation and temperature affect the climate cropping potential and set the upper limit. Based on this restriction, the Global Agro-ecological Zones (GAEZ) model is introduced here to assess the multiple cropping potential through matching both growth moisture and temperature requirements of individual suitable crops with time available for crop growth (Fischer et al., 2021). Delineation of multiple cropping zones is solely based on agro-climatic attributes calculated during GAEZ analysis. On the supply side, we introduced precipitation as a water indicator as Wu et al. (W. Wu et al., 2018) used and the following climatic characteristics for each pixel at a global scale:

a. Annual average temperature

b. Total annual precipitation

c. $TS_{t=0}$ accumulated temperature on days when mean daily temperature $\geq 0°C$.

d. $TS_{t=10}$ accumulated temperature on days when mean daily temperature $\geq 10°C$.

e. $LGP_{t=5}$ number of days with mean daily temperatures above $5°C$.

f. $LGP_{t=10}$ number of days with mean daily temperatures above $10°C$.

Tab. S4 and Tab. S5 summarize the delineation criteria for cropping potential zones in the tropics and the subtropics/temperate zones, combining the GAEZ model and precipitation.

**Cropland pressure index.** To quantify cropland pressure driven by cropland expansion in different regions (Calvin et al., 2017) of Tab. S1, an index is introduced here. The current cropland potential area for each region is aggregated spatially based on HYDE 2015 cropland area weighted by the cropping potential. Cropland pressure index is calculated by dividing the total population by cropland potential area over the regions. The current population is estimated based on populations from 2015 (Worldpop2015 (Lloyd et al., 2019)), the same year as HYDE 2015. The global gap of cropland pressure index is based on the top ten and bottom ten regions, approximately the interquartile range.

## 3 Results

**Changes in croplands and associated climates.** As shown in the History Database of the Global Environment (HYDE version 3.2) datasets for cropland distribution (Foley et al., 2005; Goldewijk et al., 2017), the global mean cropland proportion increased from 0% in 10000 BC to 28.5% in 2015 (Fig.1a). The historical development process (Fig.S12) reveals that prior to 10000 BC, there was no cropland, and vegetation coverage was entirely natural. Subsequently, croplands gradually expanded until 1850, when rapid growth ensued. By 1990, cropland expansion reached a plateau, with negligible growth observed by 2015. Consequently, we selected key time points (10000 BC, 1850, 1990, and 2015) to represent the historical evolution of

croplands. Global cropland expansion from 10000 BC to 2015 predominantly occurred in eastern America, Europe, central Africa, India, and China (Fig.1b), with notable reductions in forests, followed by declines in grasslands and shrublands (Fig.S13). From 1850 to 2015, croplands in America, Europe, Africa, and Asia continued to increase despite the encroachment of forests, grasslands, and shrublands. The rate of cropland expansion slowed due to the utilization of easily convertible land for crops and increased yield per unit area resulting from industrial fertilization, ultimately leading to decreases in some areas (Fig.S14). Between 1990 and 2015, cropland coverage declined in America, Europe, and China, attributable to expansions in forests, grasslands, and urban areas. However, significant cropland expansions persisted in the Amazon and Africa (Fig.S15).

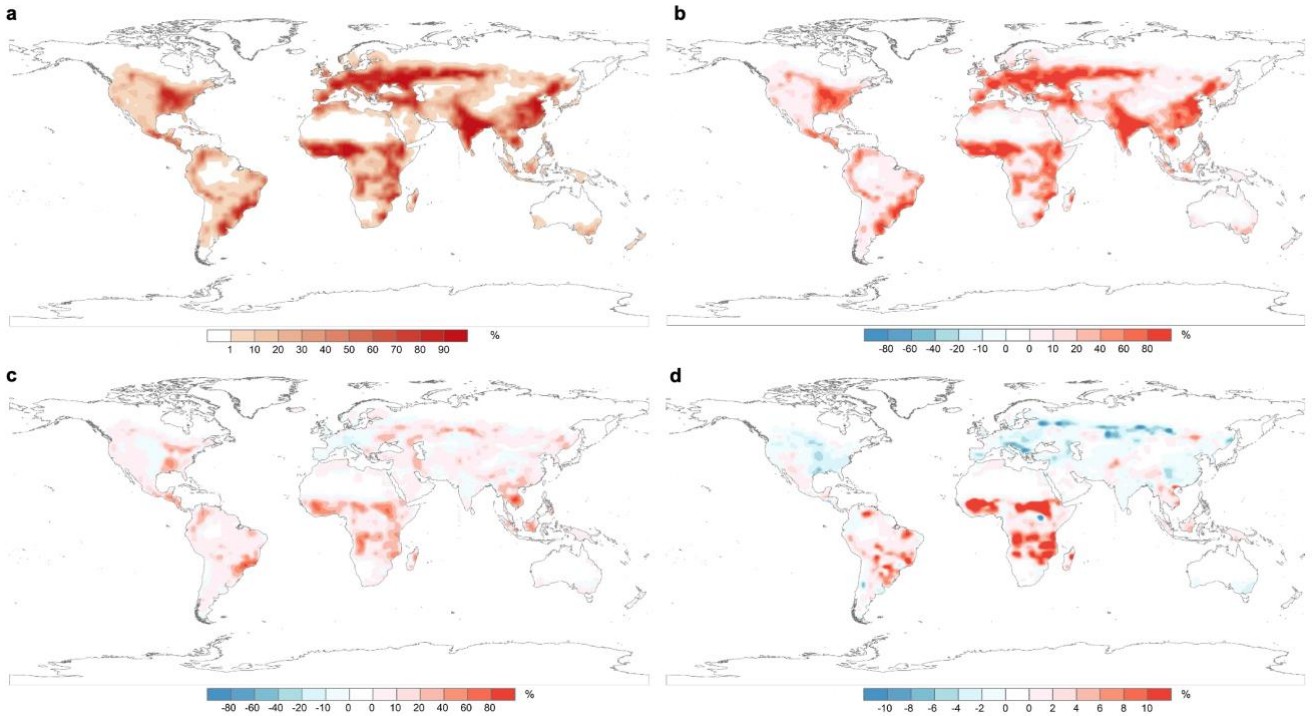

**Figure 1. Global distribution and fractional changes of cropland. a,** Cropland distribution in 2015. **b-d,** Fractional changes of cropland relative to 2015 for 10000BC (**b**), 1850 (**c**) and 1990 (**d**).

With observed land cover in 2015 and cropland-induced changes for 10000BC, 1850 and 1990 (Figs. 1 and S12-15), we performed four sets of experiments using the National Center for Atmospheric Research (NCAR) Community Earth System Model version 1.2.1 (CESM1.2.1) to obtain climates associated with the cropland states for 10000BC, 1850, 1990 and 2015, respectively (see Methods). The significant cropland expansion from 10000BC to 2015 led to notable changes in surface air temperature and precipitation (Fig. 2a&b). The pronounced warming observed over northern Eurasia primarily resulted from warm air temperature advection due to modified atmospheric circulation and increased downwelling longwave radiation. The warming further leads to increased atmospheric heat storage, as well as enhanced water vapor due to the greater atmospheric water vapor holding capacity. Therefore, the longwave downwelling flux is increased, contributing to the warming. The

warming tendency in low-latitude tropics is mainly associated with the warming effect of the reduced surface evaporation due to cropland expansion. The cooling tendency in the sub-tropics is dominated by the cool advection and decreased downwelling longwave radiation. The cooling effect over eastern North America can be ascribed to reduced surface sensible heat and decreased downwelling longwave radiation. The reduced sensible heat fluxes over eastern North America and Europe are due to the smaller surface roughness of cropland compared with the forest, enhancing aerodynamic resistance to sensible heat diffusion from the land surface to the atmosphere. Enhanced precipitation over Europe mainly arose from increased moisture advection, while diminished precipitation over India was linked to reduced moisture convergence and evaporation. The decreased precipitation over India further leads to increased concentration of absorptive aerosols such as black carbon and dust, and contributes to the warming there (Fig. S16d).

In addition to the annual surface air temperature and precipitation changes from 10000BC to 2015 (Fig. 2a&b), the temperature-derived variables also used for the climate cropping potential intensity model (see Methods) are shown in Fig. 2c-f. The annual average temperature in high latitudes of North America, Eurasia, India, South America, and central and south Africa is on the rise. At the same time, a significant cooling trend is shown in central and eastern Asia, the Middle East, North Africa, and most of the United States, resulting from the combined effects of reduced sensible heat fluxes, decreased longwave downwelling fluxes, and cool advection. $TS_{t=0}$ (accumulated temperature on days when mean daily temperature $\geq 0°C$), $TS_{t=10}$ (accumulated temperature on days when mean daily temperature $\geq 10°C$), $LGP_{t=5}$ (number of days with mean daily temperatures above $5°C$) and $LGP_{t=10}$ (number of days with mean daily temperatures above $10°C$) resemble the annual average temperature, except for LGP in the tropical zones. Similar changes for temperature and precipitation from 1850 to 2015 and 1990 to 2015 were also found (Fig.S18&19).

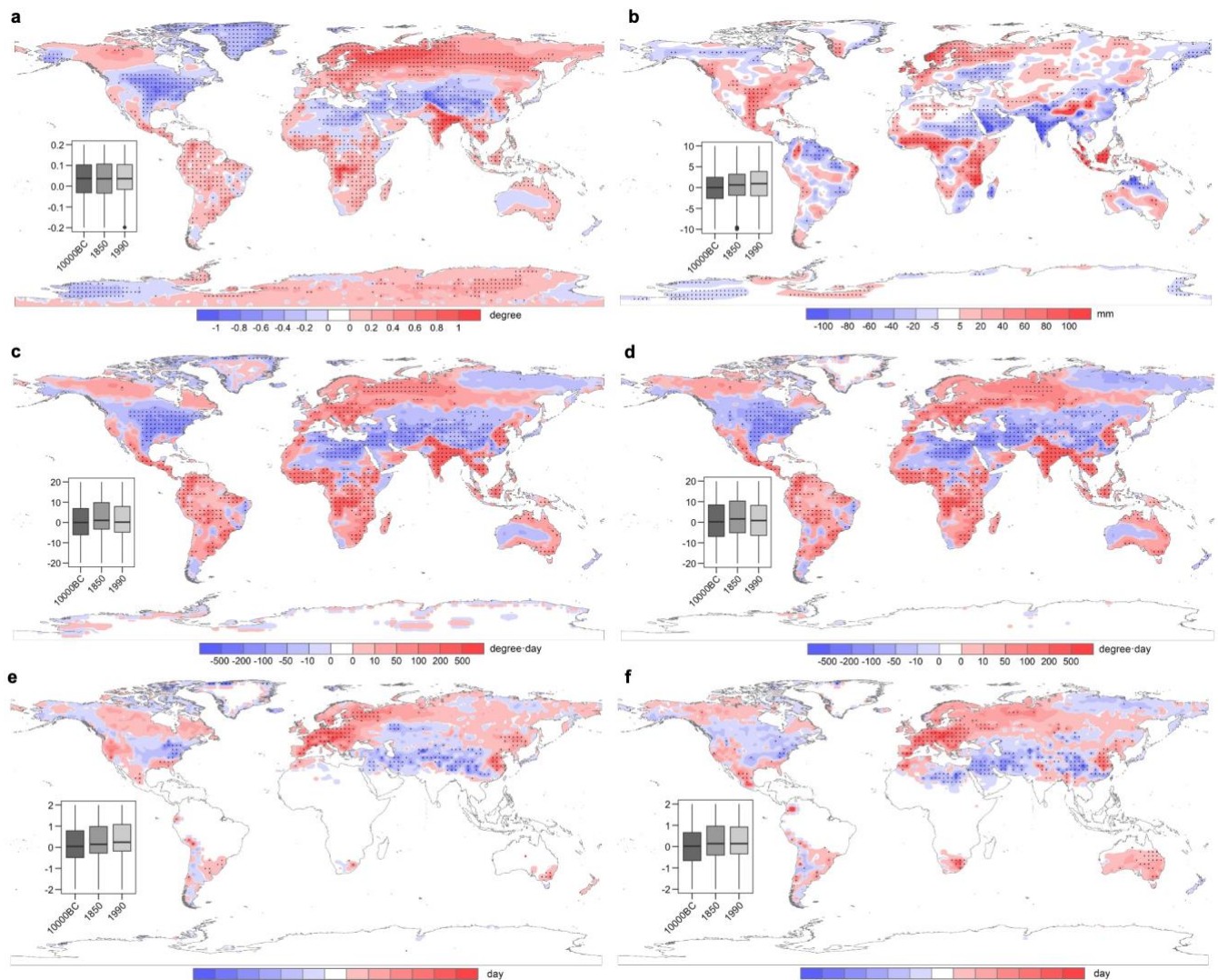

**Figure 2. Temperature and precipitation changes from 10000BC to 2015 due to cropland change. a,** Annual mean temperature. **b,** Annual mean precipitation. **c,** $TS_{t=0}$. **d,** $TS_{t=10}$. **e,** $LGP_{t=5}$. **f,** $LGP_{t=10}$. Inset box charts show the global mean value for each variable relative to 2015 for 10000BC, 1850, and 1990. The solid black dots indicate the pixels passing a two-sample t-test (95% confidence level).

**Increased cropping potential.** To explain the effects of cropland expansion on cropping potential, we use the improved Global Agro-ecological Zones (GAEZ) model to calculate the cropping potential using bias-corrected hydrothermal conditions from simulations (see Methods). Our findings reveal that global expansion from 10000 BC to the present augments the total climate cropping potential, with a cumulative value of 0.006 across the entire period (Fig. 3a). Additionally, over the time from 10000BC to 2015, the average increase in cropping potential was 0.0004 per 1% expansion of cropland, 0.0024 per 1%

expansion from 1850 to 2015, and 0.0268 per 1% expansion of cropland during 1990-2015, mirroring the slowed cropland expansion rate. A 1.2% enhancement in global cropping potential was observed within the timeframe of 10000 BC to 2015 (see inset bar chart in Fig. 3a). That is, with the development of history, the rate of cropland expansion has slowed down, but 250 the rate of increase of cropping potential caused by cropland expansion has increased significantly.

Consequently, the cropping potential attributable to the percentage of cropland expansion rises considerably. In general, cropland expansion contributes to an enhanced cropping potential, yielding a two-fold increase in crop potential. This outcome signifies an elevated upper limit for planting and increased efficacy of irrigation for fertilization through strategic cropland expansion. Hence, regulated cropland expansion could prove advantageous for croplands by amplifying cropping potential.

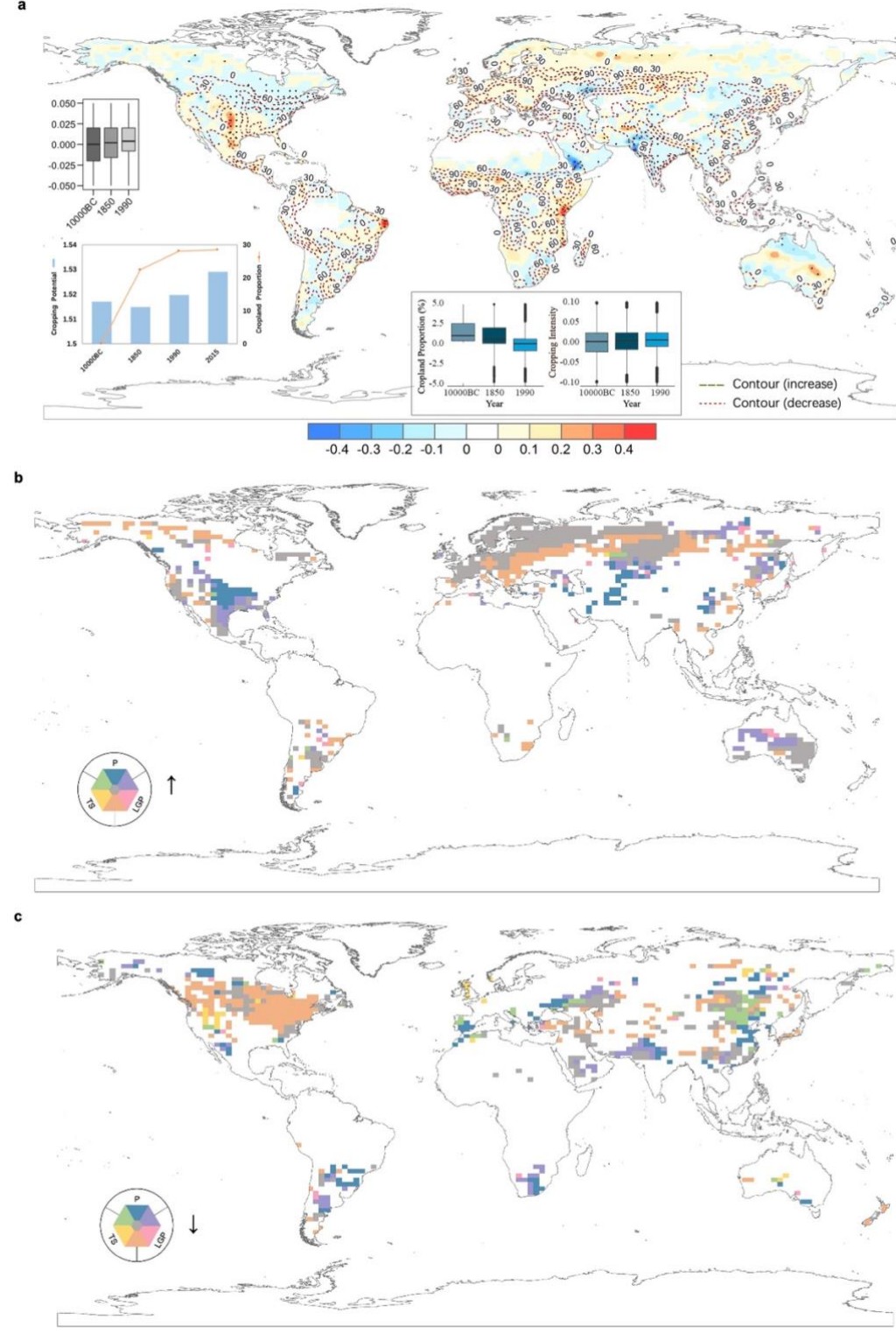

**Figure 3. Cropping potential changes and causes from 10000BC to 2015. a,** Cropping potential distribution changes superposed by the cropland expansion fraction (contours). Inset box charts show the global mean cropping potential distribution changes (%) relative to 2015 for 10000BC, 1850, and 1990, respectively. The inset bar chart represents the cropping potential from 10000BC to 2015, and the line denotes the global mean cropland fraction. The solid black dots indicate the pixels passing a two-sample t-test (95% confidence level). **b and c,** Global distributions of causes for increasing (**b**) and decreasing (**c**) cropping potential from 10000BC to 2015. Increased cropping potential caused by precipitation is shown in blue, $LGP_{t=5}$ or $LGP_{t=10}$ in pink and $TS_{t=0}$ or $TS_{t=10}$ in yellow. Combinations of the two factors are shown in: purple (precipitation, and $LGP_{t=5}$ or $LGP_{t=10}$; both factors promote the cropping potential growth); orange ($LGP_{t=5}$ or $LGP_{t=10}$, and $TS_{t=0}$ or $TS_{t=10}$); and green (precipitation, and $TS_{t=0}$ or $TS_{t=10}$). Grey indicates the increase for all the three factors: precipitation, $LGP_{t=5}$ or $LGP_{t=10}$, and $TS_{t=0}$ or $TS_{t=10}$.

From a spatial perspective, there are both upward and downward trends for cropping potential from 10000BC to 2015 (Fig. 3a). Cropland expansion leads to different cropping potential changes among separate areas. There is a tendency to decrease the cropping potential in several major agricultural areas, particularly in Asia (Central Asia, India, and China) and the northeastern United States, which belong to the temperate and subtropical east coast. The decline for central Asia and India is mainly from all the climate variables, for China is mainly from the precipitation and TS-related variables. In contrast, the temperature-related variables occupy the dominant position for the northeastern United States (Fig. 3c). While in Europe, South America, and South Africa, which belong to the tropical zones or close to the west coast zones, all agricultural areas are improving their cropping potential, caused by the increase of temperature-related variables and increased precipitation in some cases (Fig. 3b). For areas that are not currently covered by cropland (out of the contour), the Middle East and North Australia, cropland has not expanded. Their cropping potential has declined due to the decrease in temperature and precipitation affected by cropland expansion in other areas. However, in Northern Eurasia, the overall cropping potential for cropland due to climate change caused by cropland expansion is getting better, which demonstrates more favorable climatic conditions for crop growing as a result of cropland expansion elsewhere. Similar changes for temperature and precipitation from 1850 to 2015 and 1990 to 2015 were observed (Fig. S20&S21).

**Reduced cropping potential inequality.** For further analysis of copping potential inequality across the globe, an index called the cropland pressure index is introduced here. The cropland pressure index presents the pressure for cropping potential capacity (based on the cropland distribution) to carry the current population in each region (see Methods). The pressure for several leading agricultural powers is small, such as China, Russia, the United States, and India (Fig. 4). Even West Africa shows minimal pressure because of considerable cropping potential. At the same time, other regions are under great cropland pressure due to their high-density population on little cropland in places such as Japan, or their inability to do anything about the potential of future cropland, such as the Middle East. The area chart from 10000BC to 2015 is included here to quantify the inequality between regions (Fig. 4), indicating the cropland pressure index gap among the interquartile range (reduced by

290 ≈ 20). Luckily, the cropland pressure index gap between 'rich' and 'poor' regions has narrowed, and inter-region trade, such as the global trade in agricultural products, also narrows the gap.

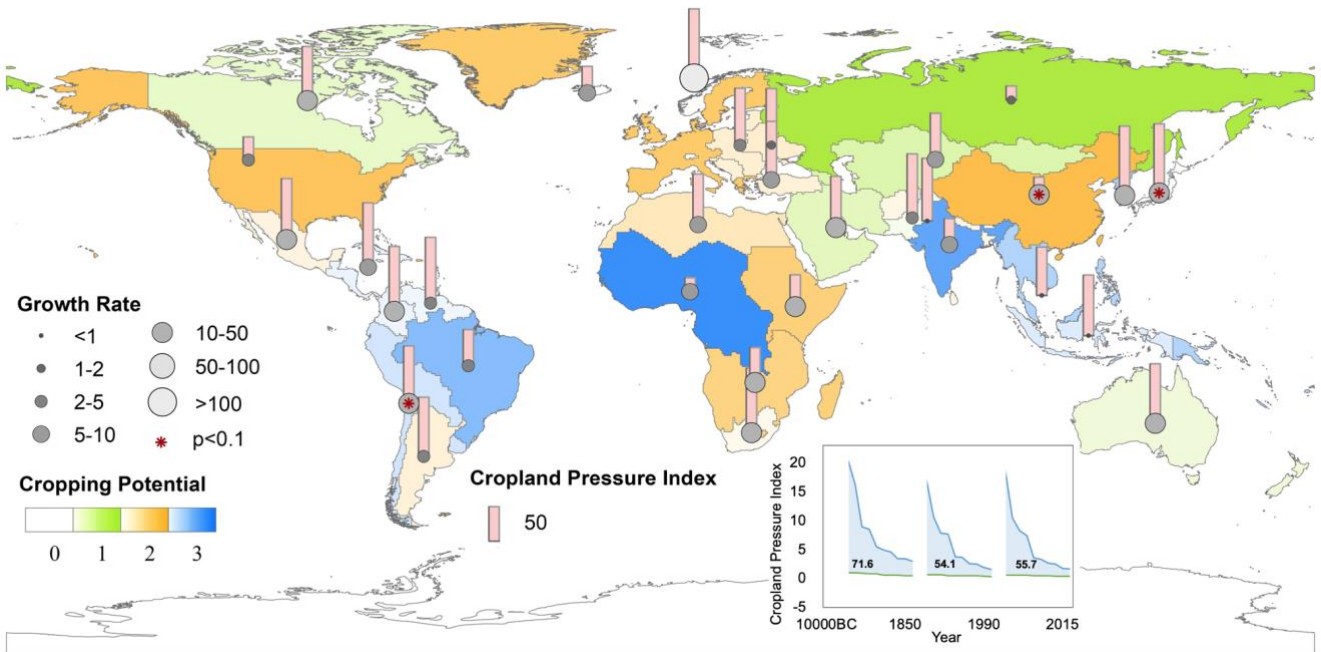

**Figure 4. Cropping potential inequality at regional scale from 10000BC to 2015.** The background color represents the cropping potential value in 2015, with the transparency indicating the cropland fraction in this region: The darker the color, 295 the larger the fraction of cropland. The grey circle shows the cropping potential growth rate (× 10000) from 10000BC to 2015, and the * in red means the region passed the significance test of 90%. The red bar and the area chart display the crop pressure index in 2015 and the top ten and bottom ten regions from 10000BC to 2015. In the inner plot, the x-axis represents the countries sort the top ten of the 'rich' (green line) and 'poor' (blue line) cropland pressure index. The number in the blue area displays the cropland pressure index gap between 'rich' and 'poor' regions.

From a historical development perspective, as the growth rate from 10000BC to 2015 is shown in Fig. 4, the bigger and whiter the circle is, the faster the cropping potential growth rate, which means a huge potential for further cropland expansion. The figure shows that it nearly follows the trend of decreasing cropping potential growth from high latitudes to low latitudes (Chen, 2018; IPCC, 2014; Zhang & Ma, 2019). The region with the fastest growth potential is North Europe (European Free 305 in Tab. S1). The minimum potential growth rate is in Southeast Asia, indicating the higher latitudes have greater potential for improving hydrothermal conditions. The cropping potential growth rate in Africa, East Asia, and some Latin American countries is relatively rapid worldwide, with China, Japan, and Argentina passing the significance test, revealing the huge cropping potential of cropland expansion for these regions in the future. As for those regions with lower growth rates, the cropland expansion might be a low efficient way to increase the crop potential yield.

**4 Conclusion**

In conclusion, the study indicates that spanning from 10,000 BC to 2015, significant expansion of cropland has led to notable changes in near-surface temperature and precipitation patterns. Notably, average annual temperatures in high latitudes across regions like North America, northern Eurasia, India, South America, and central and southern Africa have been on the

315 rise. Conversely, central and eastern Asia, the Middle East, northern Africa, and much of the United States have experienced a distinct cooling trend. Moreover, total annual precipitation has notably decreased in subtropical regions such as South China, Thailand, India, the Middle East, and northern Africa, while tropical regions like East Africa, Central Africa, and Southeast Asia have predominantly seen precipitation increases. These varied climate shifts have resulted in disparities in cropping potential across different regions.

Several major agricultural zones, notably in Central Asia, India, China, and northeastern United States, have observed a decline in cropping potential. However, agricultural regions such as Europe, South America, and South Africa, where cropland is expanding, have witnessed an increase in cropping potential. Additionally, the growth rate of cropping potential generally follows a pattern of diminishing rates from high latitudes to low latitudes, indicating greater cropping potential for improving hydrothermal conditions in higher latitudes. In regions with low potential growth rates of cropland, expanding cropland has

minimal impact on enhancing local potential production.

When assessing current cropland pressure based on cropping potential, several major agricultural countries face minimal pressure, while other regions struggle with significant pressure due to population density on limited cropland or challenges in accessing cropping potential. Nevertheless, a positive observation is that the global disparity in cropland pressure has decreased, reflecting a reduction in inequality in cropping potential among regions. Overall, cropland expansion has led to an

330 increase in cropping potential, highlighting the dual benefits of global cropland expansion for total crop production.

However, it is worth noting that all information presented is based on climate cropping potential and experiments are conducted using only one climate model, which may be insufficient. Specific circumstances must also be determined based on factors such as soil conditions and altitude when implemented on the ground. Additionally, significant changes in cropping potential are observed only in prominent climate change regions globally. This study reveals the overall trend conclusion from

335 a global perspective, and the accurate benefits brought by cropland expansion need to be analysed based on specific regional conditions.

**Acknowledgments**

This research is funded by the National Key R&D Program of China (grant number: 2019YFA0606601), Tsinghua University
Initiative Scientific Research Program (grant number: 20223080017), the National Key Scientific and Technological Infrastructure project "Earth System Science Numerical Simulator Facility" (EarthLab), and the Laboratory fund of Chinese Academy of Sciences (grant number: CXJJ-22S032).

**Code availability**

The     CESM1.2.1     source     code     can     be     downloaded     from     the     CESM     official     website https://www.cesm.ucar.edu/models/cesm1.2/.

**Data availability**

The HYDE cropland distribution data set is available through Goldewijk et al. 2017. The CESM simulation data is available
from the figshare link:
https://figshare.com/s/b3311a5486d653bb4e43.

**Additional Information**

Supplementary information (figures and tables) is in a separate file.

**Author Contributions**

Xiaoxuan Liu drafted the manuscript with contributions from all the authors. Xiaoxuan Liu and Shu Liu processed the data and finished the data analysis. Le Yu and Yong Wang designed and instructed the study. All authors participated in revising the paper.

**Competing Interests**

The authors declare no competing interests.

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
