# Peer review of "Global Cropland Expansion Enhances Cropping Potential and Reduce its Inequality among Countries"

_Earth System Dynamics, 2023_

## Author Comment (AC1)

**Responses to the comments of RC1**

**Comment 1:** In the AGCM simulations with different cropland conditions, how are the external forcings given (e.g., at the 1850 or 2015 level)? Will the use of different external forcing levels (e.g., at the 1850 or 2015 level) affects the climate response? This is related to the nonlinear responses of climate to land use / land cover change and external forcings such as greenhouse gases and aerosols. At least some discussions on this issue would be helpful.

Reply: *As we focus on the climatic effects of the external forcing of cropland expansion, in the AGCM simulations with different cropland conditions, other external forcings such as anthropogenic aerosol emissions and greenhouse gas concentrations were all fixed in the present-day climatological conditions (1982–2001 mean) (Eyring et al., 2016). However, as mentioned by the reviewer, due to the nonlinear responses of climate to external forcings, the use of different external forcing levels may affect climate responses. Associated discussions have been added in revised manuscript:*

*Added in manuscript: Taking into account the nonlinear responses of climate to external forcing (Rohrschneider, Stevens, & Mauritsen, 2019), employing various external forcing levels such as 10000 BC, 1850, 1990, and 2015 would inevitably influence the climate's reaction. Thus, we ran the Atmospheric Model Intercomparison Project (AMIP)-type (Eyring et al., 2016) experiments, using fully prognostic atmosphere and land models with prescribed, seasonally varying present-day climatological (1981-2001 mean) sea surface temperatures and sea ice concentrations (Hurrell et al., 2013).*

**Comment 2:** L130-145: The forms of Equations (1) and (2) do not seem correct.

*Reply: Sorry for uploading the wrong equations due to DOC version issues. We have corrected them as following and also in revised manuscript.*

$$\delta \bar{T} \approx \gamma^{-1} \left( -\delta \overline{(\vec{V_h} \cdot \nabla_h T)} + \delta \overline{(S_p \omega)} + \delta \overline{Q_s} + \delta \overline{Q_{ld}} + \delta \overline{F_{sh}} + \delta \overline{Q_q} \right) \quad (1)$$

$$\delta \bar{P} \approx \delta \overline{(-W \nabla \cdot \vec{V})} + \delta \overline{(-\vec{V} \cdot \nabla W)} + \delta \bar{E} \quad (2)$$

**Comment 3:** Fig. 3a: The inset blue bars (global mean cropping potential) show no change from 10000BC to 2015. Please check if there is an error. In addition, as the map shows, only very limited locations show significant change in cropping potential (indicated by solid black dots) from 10000BC to 2015. This is also seen from Fig. 4. So I wonder if the conclusion that "a 28% increase in cropland expansion has led to a 1.2% enhancement in global cropping potential" (L24) is overstated. The low significance needs to be clarified.

*Reply: Actually, the variation of average cropping intensity worldwide is relatively small. And due to the fluctuation in cropping intensity worldwide, calculating global averages may dilute local variations. While individual bar charts exhibit some variability, the representation on the graph becomes limited when error bars are added. To avoid ambiguity, we have adjusted the scale of the statistical graph for better representing changes between years. And we added two alternative methods as follows that more directly illustrate changes in cropping intensity between different years, rather than focusing on the absolute values each year.*

[Figure]

*Also, summing up all the pixels in Figure gives the global average cropland potential value added from 10,000 BC to 2015 being 0.012. To prevent ambiguity, we added the analysis of low significance of 90% and 85% confidence level, the significant change in cropping potential (indicated by solid black dots) from 10000BC to 2015 could be seen (We added Fig.S22 in Supplementary Information). And we also added a note to warn of the limitations of 95% significance for our current conclusions in revised manuscript.*

*Added in manuscript: Additionally, significant changes in cropping potential are observed only in prominent climate change regions globally. This study reveals the overall trend conclusion from a global perspective, and the accurate benefits brought by cropland expansion need to be analysed based on specific regional conditions.*

[Figure]

**Fig. S22.** Cropping potential changes with (a)95%, (b)90%, (c)85%, (d)80%confidence level from 10000BC to 2015.

**Comment 4:** L275-278 and Fig. 4: It is indicated that cropping potential growth rate is greater in high latitudes than in low latitudes. I wonder if this statement is robust. This can be confirmed by investigating the relationship between cropping potential growth rate and latitude.

*Reply: From Fig.4, we could find the cropping potential growth rate is greater in high latitudes than in low latitudes. However, it is not an alone result, since some previous research has found similar conclusions: Chen et al. found the zones with a higher latitude show possible increased multiple cropping intensity (Chen, 2018); IPCC reported that climatic warming could potentially benefit agriculture at higher latitudes by promoting greater cropping intensity (IPCC, 2014); And climate*

*change could transform cropland from single cropping to multiple cropping at higher latitudes (Zhang & Ma, 2019). Also, we have added some references as the reviewer suggested in revised manuscript:*

*Added in manuscript: The figure shows that it nearly follows the trend of decreasing cropping potential growth from high latitudes to low latitudes (Chen, 2018; IPCC, 2014; Zhang & Ma, 2019).*

**Comment 5:** Fig. 4: The inset plot about the cropland pressure index between "rich" and "poor" regions needs further clarification (e.g., what the x-axis is).

*Reply: The x-axis is the countries sort the top ten of the 'rich' (green line) and 'poor' (blue line) cropland pressure index. The number in the blue area is the cropland pressure index gap between 'rich' and 'poor' regions. Also, we have added the axis description in Fig.4 of revised manuscript.*

*Added in manuscript: In the inner plot, the x-axis represents the countries sort the top ten of the 'rich' (green line) and 'poor' (blue line) cropland pressure index. The number in the blue area displays the cropland pressure index gap between 'rich' and 'poor' regions.*

**Comment 6:** The analyses are based on only one model, which weakens the conclusion overall. This limitation needs to be acknowledged. In addition, while the climatological mean bias in temperature and precipitation has been corrected, a detailed model evaluation including the spatial distributions and PDFs of temperature and precipitation is needed, to provide basic information about the reliability of the results.

*Reply: Previous studies have evaluated CESM, demonstrating its reliability and accuracy in simulating climate change. In this paper, we provide reasons for using CESM for simulations and include relevant references to underscore the reliability of analyses based on CESM experimental results (Hurrell et al., 2013; Kay et al., 2015).*

*Added in manuscript: The reliability of the CESM has been confirmed in numerous previous studies, making it suitable for applications such as climate change simulation and climate model analysis (Hurrell et al., 2013; Kay et al., 2015).*

*Moreover, as suggested by reviewers, relying solely on one model has limitations. Therefore, we have added explanations regarding these limitations. In the future, multi-model climate simulation can help to provide a better understanding of the climatic effects of cropland expansion.*

*Added in manuscript: However, it is worth noting that all information presented is based on climate cropping potential and experiments are conducted using only one climate model, which may lead to biased outcomes. Specific circumstances must also be determined based on factors such as soil conditions and altitude when implemented on the ground.*

*For the temperature and precipitation, we used the widely applied 'delta method' for climate model bias correction with AgERA5. For the detailed model evaluation, we added the spatial distributions and PDFs of temperature and precipitation as suggested. The spatial distributions and PDFs results are shown in the new Fig. S11.*

*Added in manuscript: We use the spatial distributions and PDF (Probability Density Function) to verify the correctness of the corrected temperature and precipitation. The PDFs results and the Bias-corrected temperature with precipitation based on AgERA5 are shown in Fig. S8-11.*

[Figure]

**Fig. S11.** Bias-corrected T and P in 2015 based on ERA5. (a) uncorrected temperature (b) uncorrect precipitation (c) corrected temperature (d) corrected precipitation (e) PDF of    corrected temperature (f) PDF of corrected precipitation (g) ERA5 temperature (h) ERA5 precipitation (i) PDF of ERA5 temperature (j) PDF of ERA5 precipitation.

*References:*

- *Eyring, V., Bony, S., Meehl, G. A., Senior, C. A., Stevens, B., Stouffer, R. J., & Taylor, K. E. (2016). Overview of the Coupled Model Intercomparison Project Phase 6 (CMIP6) experimental design and organization. Geoscientific Model Development, 9(5), 1937–1958.*

- *Rohrschneider, T., Stevens, B., & Mauritsen, T. (2019). On simple representations of the climate response to external radiative forcing. Climate Dynamics, 53, 3131-3145.*

- *Hurrell, J. W., Holland, M. M., Gent, P. R., Ghan, S., Kay, J. E., Kushner, P. J., . . . Lindsay, K. (2013). The community earth system model: a framework for collaborative research. Bulletin of the American Meteorological Society, 94(9), 1339-1360.*

- *Chen, B. (2018). Globally increased crop growth and cropping intensity from the long-term satellite-based observations. ISPRS Annals of the Photogrammetry, Remote Sensing and Spatial Information Sciences, 4, 45-52.*

- *IPCC, I. (2014). Climate change 2014: Synthesis report. Contribution of working groups I, II and III to the fifth assessment report of the intergovernmental panel on climate change.*

- *Zhang, X., & Ma, X. (2019). Misplaced optimism in agricultural land usage driven by newly available climate resources: A case study of estimated and realized cropping intensity in northern and northeastern China. Climate Risk Management, 25, 100194.*

- *Kay, J. E., Deser, C., Phillips, A., Mai, A., Hannay, C., Strand, G., et al. (2015). The Community Earth System Model (CESM) Large Ensemble Project: A Community Resource for Studying Climate Change in the Presence of Internal Climate Variability. Bulletin of the American Meteorological Society, 96(8), 1333–1349. https://doi.org/10.1175/bams-d-13-00255.1*

---

## Author Comment (AC2)

**Responses to the comments of RC2**

**Comment 1:** The authors motivate the study by describing that the underlying mechanisms behind the effect of cropland expansion on cropping potential remain unexplored (line 23). However, there is very little discussion about the underlying mechanisms. The simulated results by the earth system model are directly used as a forcing to cropping potential model and the assessment about the change in cropping potential is done. I would suggest to add relevant discussion about following points.

**C1.1:** The tropics and sub-tropics seem to show a clear tendency where wet regions get warmer and dry regions get cooler as a result of cropland expansion. Some discussion should be added for why this could be the case?

*Reply:* *The warming tendency in low-latitude tropics (mentioned as wet regions by the reviewer) is mainly associated with the warming effect of the reduced surface evaporation due to cropland expansion (Fig. S16f). The sub-tropics shows less cropland expansion. Therefore, the cooling tendency in the sub-tropics (mentioned as dry regions by the reviewer) are dominated by large-scale atmospheric adjustments such as the cool advection (Fig. S16a). The local cooling effect of cropland expansion by reducing sensible heat fluxes weakens the upper-troposphere westerly jet and thus results in cool advection in the sub-tropics. The cooling effect leads to decreased atmospheric heat storage, as well as decreased water vapor due to the reduced atmospheric water vapor holding capacity (saturation vapour pressure). Therefore, the longwave downwelling flux is decreased, contributing to the cooling in the sub-tropics (Fig. S16e). We added some discussion in Section 3 Results of revised manuscript.*

*Added in manuscript: Additionally, the increase in saturation vapour pressure enhancing water vapor led to an increase in longwave downwelling flux. The warming tendency in tropics and subtropics is mainly associated with the warming effect of the reduced surface evaporation due to cropland expansion.*

**C1.2:** The warming in tropics (e.g India) can-not be attributed to changes in latent heat flux alone (line 200). Figure S16 clearly shows a decrease in solar radiative heating which primarily reflects decrease in cloud cover. The effect of deforestation on cloud-cover have already been emphasized (Duveiller et al 2021) and cloud-radiative effects have been shown to significantly affect surface temperatures (Ghausi et al., 2023).

*Reply:* *The adjustment of solar radiation in response to cloud cover changes directly affects surface temperature, and then indirectly influences near-surface air temperature via the near-surface turbulence of sensible and latent heat (Fig. S16c&f) (Ghausi et al., 2023). Here, the shortwave radiative heating on near-surface air temperature results from reduced precipitation which leads to the increased concentration of absorptive aerosols such as black carbon and dust (Liu et al., 2019). We added the description in the revised manuscript.*

*Added in manuscript: Warming in the tropics and subtropics was governed by changes in solar radiation affected by cloud cover and latent heat flux stemming from deforestation (Ghausi, Tian, Zehe, & Kleidon, 2023).*

**C1.3:** About warming of northern Eurasia: Changes in warm-air temperature advection mostly results in increased downwelling longwave radiation (Rld) as result of increase in atmospheric heat storage. Therefore, these are confounding effects and not really the independent of each other (See also Tian et al., 2022; Tian et al., 2023). This point should be made clear.

*Reply:* *The warming of northern Eurasia due to the warm advection leads to increased atmospheric heat storage, as well as enhanced water vapor due to the greater atmospheric water vapor holding*

*capacity (saturation vapour pressure). Therefore, the longwave downwelling flux is increased, contributing to the warming. The positive feedback has been additionally discussed in the revised manuscript.*

*Added in manuscript: Additionally, the increase in saturation vapour pressure enhancing water vapor led to an increase in longwave downwelling flux.*

**C1.4:** Over Europe, the sensible heat flux has reduced substantially. Do authors have an explanation for that?

*Reply: The surface roughness of cropland is smaller than that of the forest. Therefore, the expansion of cropland with decreased forest cover enhances aerodynamic resistance to sensible heat diffusion from the land surface to the atmosphere, leading to cooling effects (Fig. S16c).*

**C1.5:** Line 208: There is no discussion about the factors that cause the cooling trend over these regions.

*Reply: The significant cooling in central and eastern Asia, the Middle East, North Africa, and most of the United States primarily results from the combined effects of reduced sensible heat fluxes, decreased longwave downwelling fluxes, and cool advection. We added in the revised manuscript.*

**C1.6:** Other biophysical factors that can mediate changes in temperatures in response to changing vegetation type should be discussed (Kleidon et al., 1998; Lee et al., 2011; Chen et al., 2020).

*Reply: The contributions of surface sensible and latent heat fluxes to air temperature changes involve the impacts of many biophysical factors, such as surface albedo, surface emissivity, aerodynamic resistance, and surface resistance (Lee et al., 2011; Chen et al., 2020). This is because these biophysical factors directly affect surface temperature, and then indirectly influence near-surface air temperature by the near-surface turbulence of sensible and latent heat (Zeng et al., 2017; Li et al., 2019). The discussion has been added in the revised manuscript.*

*Added in manuscript: At the same time, a significant cooling trend is shown in central and eastern Asia, the Middle East, North Africa, and most of the United States, resulting from the combined effects of reduced sensible heat fluxes, decreased longwave downwelling fluxes, and cool advection.*

**C2:** Line 223: The cropping potential is described using a cumulative value. Is it the cropland pressure index discussed in lines (170). Does it have a dimension? There should be some information provided to interpret its values.

*Reply: The cropping potential value is not the cropland pressure index. The cropping potential, in this article is climate cropping potential, denotes the utmost capacity for multi-cropping achievable after thorough climate resource assessment. While the cropland pressure index presents the pressure for cropping potential capacity (based on the cropland distribution) to carry the current population in each region, which is calculated by dividing the total population by cropland potential area over the regions. To prevent ambiguity, we have added a conceptual explanation of the cropping potential. The explanation of cropland pressure has been added in revised manuscript.*

*Added in manuscript: Climate cropping potential denotes the utmost capacity for multi-cropping achievable after thorough climate resource assessment.*

*Added in manuscript: Cropland pressure index is calculated by dividing the total population by cropland potential area over the regions. The cropland pressure index presents the pressure for cropping potential capacity (based on the cropland distribution) to carry the current population in each region.*

**C3:** Figure 3 b,c: How are changes in cropping potential attributed to individual variables? To what extent are the 5 variables (describing temperatures) correlated?

*Reply: Here we categorize the variables used to calculate cropping potential into three groups: P (precipitation), LGP (number of days), and TS (accumulated temperature). Employing Bottleneck analysis, a systematic approach aimed at identifying primary limiting factors, we determine which types of variables influence the overall increase in cropping potential. Furthermore, this attribution is not limited to individual variables, but considers 1 (precipitation), or 2 (TS or LGP), or 3 (green or purple), and even 6 (gray) variables.*

*Regarding the correlation among temperature description variables, although these variables are all temperature-based, they vary in dimensions. We introduced these variables because GAEZ specifies their necessity for calculating cropping potential (Fischer et al., 2021). As depicted in the Figure 3, most of the temperature-related attribution results are depicted in orange (representing all temperature description variables). Therefore, bottleneck analysis is conducted solely based on GAEZ variables, which does not imply a lack of correlation among them.*

**Minor:**

**M1:** There seems to be some typesetting error in equations 1 and 2.

*Reply: Sorry for uploading the wrong equations due to DOC version issues. We have corrected them as following and also in revised manuscript.*

$$\delta\bar{T} \approx \gamma^{-1}\left(-\delta\overline{(\vec{V_h}\cdot\nabla_h T)} + \delta\overline{(S_p\omega)} + \delta\overline{Q_s} + \delta\overline{Q_{ld}} + \delta\overline{F_{sh}} + \delta\overline{Q_q}\right) \quad (1)$$

$$\delta\bar{P} \approx \overline{\delta(-W\nabla\cdot\vec{V})} + \overline{\delta(-\vec{V}\cdot\nabla W)} + \delta\bar{E} \quad (2)$$

**M2:** The legend size in figure 3b and 3c should be increased.

*Reply: We have enlarged the figure b and c in Fig.3, Fig.S20 and Fig.S21 as suggested.*

*References:*
- *Ghausi, S. A., Tian, Y., Zehe, E., & Kleidon, A. (2023). Radiative controls by clouds and thermodynamics shape surface temperatures and turbulent fluxes over land. Proceedings of the National Academy of Sciences, 120(29), e2220400120.*
- *Liu, S., Liu, X., Yu, L., Wang, Y., Zhang, G. J., Gong, P., et al. (2021). Climate response to introduction of the ESA CCI land cover data to the NCAR CESM. Climate Dynamics, 1-19.*
- *Lee, X., Goulden, M., Hollinger, D. et al. Observed increase in local cooling effect of deforestation at higher latitudes. Nature 479, 384–387. https://doi.org/10.1038/nature10588 (2011).*
- *Chen, C., Li, D., Li, Y., Piao, S., Wang, X., Huang, M., Gentine, P., Nemani, R.R. and Myneni, R.B., Biophysical impacts of Earth greening largely controlled by aerodynamic resistance. Science advances, 6(47), p.eabb1981 (2020).*
- *Zeng, Z., Piao, S., Li, L. Z. X., Zhou, L., Ciais, P., Wang, T., et al. (2017). Climate mitigation from vegetation biophysical feedbacks during the past three decades. Nature Climate Change, 7(6), 432–436.*
- *Li, Y., Piao, S., Chen, A., Ciais, P., & Li, L. Z. X. (2019). Local and teleconnected temperature effects of afforestation and vegetation greening in China. National Science Review, 7(5), 897–912.*

- *Fischer, G., Nachtergaele, F. O., van Velthuizen, H., Chiozza, F., et al.(2021). Global Agro-ecological Zones (GAEZ v4)-Model Documentation.*

---

## Author Response (AR2)

**Responses to the comments of Editor**

**Comment 1:** Your reference list includes works "in review". Such works can be cited upon submission if being available to the reviewers. They should not be cited in the final, accepted manuscript, unless published, accepted for publication, or available as preprint with a DOI. With the next file upload request, please provide the text for the Short summary in English language and make sure not to exceed 500 characters incl. spaces.

*Reply: We replaced the reference that are accepted with the review one (See Line 68, 510-511). And we provide the short summary in English: An increase of 28% in cropland expansion since 10,000 BCE has led to a 1.2% enhancement in the global cropping potential, with varying efficiencies across regions. The continuous expansion has altered the support for population growth and has had impacts on climate and biodiversity, highlighting the effects of climate change. It also points out the limitations of previous studies.*